# Active Meta-Device for Dual-Transmission Windows with Tunable Angular Dispersion Characteristics

**DOI:** 10.3390/ma15103686

**Published:** 2022-05-20

**Authors:** Chenchen Li, Hui Bai, Mingbao Yan, He Wang, Zhiqiang Li, Wenjie Wang, Jiafu Wang, Shaobo Qu

**Affiliations:** 1Department of Basic Sciences, Air Force Engineering University, Xi’an 710051, China; lcc1398286013@126.com (C.L.); baihui9704@163.com (H.B.); 18066540235@163.com (H.W.); qiangzhilee@hotmail.com (Z.L.); wwjie@163.com (W.W.); wangjiafu1981@126.com (J.W.); qushaobo@mail.xjtu.edu.cn (S.Q.); 2Shannxi Key Laboratory of Artificially-Structured Functional Materials and Devices, Xi’an 710051, China

**Keywords:** active meta-atom, tunable angular dispersion, dual-transmission windows

## Abstract

Tailoring electromagnetic properties by meta-devices has aroused great interest with respect to manipulating light. However, the uncertainty of angular dispersion introduced by the incident waves prevents their further applications. Here, we propose a general paradigm for achieving dual-transmission windows while simultaneously eliminating the corresponding angular dispersions by a dynamic manner. The strategy of loading varactor diodes into a plasmonic meta-atom is used. In this way, the blue shifts of angular dispersion can be dynamically compensated by the red shifts introduced by the varactor diodes when driven by bias voltage. As a proof-of-principle, an active meta-atom with varactor diodes is presented. The varactor diodes embedded can independently regulate dual-transmission windows. The test results are consistent with the simulation ones. The presented meta-device is used for intelligent radome, angle-multiplexed communications, and incident-angle-insensitive equipment while providing tunable angular dispersion properties.

## 1. Introduction

In the past few decades, the emerging metamaterials have been intensively researched and developed in the electromagnetic (EM) community [1]. Metasurfaces, the 2D equivalent of metamaterials, are artificial subwavelength structures that regulate electromagnetic (EM) waves [2]. The emergence of 4D technology has led to a closer relationship between structure and performance [3]. They can achieve unique characteristics that can hardly be found in natural materials. They can be applied in microwave, acoustic, and visible regions, including vortex beams [4,5], light refraction [6,7], and invisibility cloaks [8]. Under normal incidence conditions, most of these modulation effects can be achieved [9,10]. However, as the angle of incidence increases, the angular dispersion inevitably affects the EM properties of the metasurface. Nowadays, active metasurfaces [11,12] have provided a dynamic and flexible approach to achieve electromagnetic regulation. Active components [13,14] or phase-change materials [15,16] have been introduced to achieve reconfigurable properties. They are more flexible and diverse in efficiently regulating the amplitude, phase, and polarization of EM waves [17,18]. In 2014, a programmable metasurface was presented to modulate EM waves in a programmable way [19]. The functional regulation of the far field and near field can be provided by simply changing the digital coding sequences [20,21,22].

In different application scenarios, the angular dispersion is a vital issue that requires solving. One of the angular dispersions for transmission windows is the resonant frequency shift, which will influence transmission characteristics. With the rapid development of modern communication and antenna systems, the meta-device of dual-transmission windows without angular dispersion is urgently required. Wide-angle meta-devices [23,24] and angle-multiplexed meta-devices [25,26,27] are designed to eliminate the effect of angular dispersion. However, they were obtained through brute-force simulation. In [28], the angular dispersions in metasurfaces were revealed by near-field couplings (NFCs) among adjacent meta-atoms, but the experimental demonstrations were limited to low-frequency domains. The underlying mechanisms of angular dispersions still need to be further explored and developed. By introducing active components, active metasurfaces can achieve multifunctional and continuous modulation of EM waves, including amplitude, phase, and polarization. In this way, active metasurfaces possess the great potential to become an adequate candidate to eliminate the effect of angular dispersion in a dynamic way.

In this paper, a general paradigm for tunable angular dispersion with active components is proposed. The method of loading the varactor diodes into passive meta-atom is employed to tune the angular dispersion. With the increase of the incident angle, the blue shifts of the angular dispersion appear. Simultaneously, the red shifts are introduced by the varactor diodes when driven by bias voltage. Red shifts and blue shifts counteract each other to achieve angular dispersion compensation. A triple-layer meta-device with active components is proposed as a proof-of-principle. Varactor diodes are, respectively, embedded on the top and bottom layer. They can independently regulate the resonant frequency of the transmission window. According to the simulation and test results, the proposed structure provides a dynamic method for regulating the angular dispersion. This makes it a great prospect for applications in incident-angle-insensitive meta-devices and angle-multiplexed communications.

## 2. Underlying Mechanism

The meta-device can be regarded as a combination of various impedances by the equivalent circuit model (ECM). The ECM provides a method to achieve tunable angular dispersion by adjusting the equivalent impedance.

As shown in Figure 1, when the EM wave is incident, the meta-device divides it into two parts: the reflected EM wave and the transmitted EM wave. Here, the frequency of the transmitted EM wave is determined by the equivalent impedance of the meta-device. According to transmission line theory [29], the reflection and transmission coefficients of the equivalent circuit are calculated from Equations (1) and (2):(1)$$\left|S_{11}\right|=\left|\frac{Z_{inr}-Z_{0}}{Z_{inr}+Z_{0}}\right|,$$
(2)$$\left|S_{21}\right|=\frac{2}{\left|\left(2+\frac{Z}{Z_{1}}+\frac{Z}{Z_{2}}\right)\cos\beta t+j\left(2+\frac{ZZ_{1}+ZZ_{2}+Z^{2}}{Z_{1}Z_{2}}\right)\sin\beta t\right|},$$
with
$$Z_{inr}=\frac{Z_{f2}\cdot Z_{1}}{Z_{f2}+Z_{1}},\;Z_{f2}=Z\frac{Z_{f1}+jZ\tan\beta t}{Z+jZ_{f1}\tan\beta t},\;Z_{f1}=\frac{Z_{2}\cdot Z_{0}}{Z_{2}+Z_{0}},Z_{0}=\sqrt{\frac{\mu_{0}}{\varepsilon_{0}}}\approx 377\Omega ,$$
and
$$\beta =\frac{2\pi f}{c_{0}}.$$

*Z*_0_, *Z*_1_, *Z*_2_ and *Z* are the equivalent impedances of the air on both sides, the top and bottom metal patterns of the meta-device, and the substrate, respectively. The c_0_ represents the speed of light in vacuum and t is the time for EM wave to pass through dielectric substrate.

When |*S*_21_| = 1 and |*S*_11_| = 0, *f* is regarded as the resonant frequency of the transmission window. It can be clearly seen that the resonant frequency *f* is determined by the equivalent impedance of the meta-device. The corresponding impedance of ECM changes differently in both TE and TM modes when the angle of incidence changes to *θ*. In the TE mode, free-space impedance *Z*_0_ is changed to *Z*_0_/cos*θ* and the impedance of the dielectric substrate *Z* becomes *Z*/cos*θ*. Conversely, the impedances of free-space (*Z*_0_) and the dielectric substrate (*Z*) are changed to Z_0_·cos*θ* and *Z*·cos*θ* in the TM mode. In addition, the equivalent impedances *Z*_1_, *Z*_2_ of the top and bottom layer also change based on the arrangement of the specific metallic pattern. When the incident angle *θ* increases, the impedance of the equivalent circuit varies leading to a shift in the resonant frequency *f*, which is known as the angular dispersion. Therefore, the angular dispersion can be tunable by introducing active components to diversely adjust the impedance of the ECM.

## 3. Design and Analysis of the Meta-Device

As shown in Figure 2, the proposed meta-atom is composed of three metallic layers with two thin substrates in between. The substrate used is F4B. Its relative permittivity and loss tangent angle are 2.65 and 0.001, respectively. The larger and smaller split square rings are placed on the top and bottom layers, respectively. The structure of the middle layer is a metal grid, which is equated to the inductance *L*_m_. It is mutually coupled with the metallic structures on the top and bottom layers and form two parallel L-C resonances. Therefore, the structure of dual-transmission windows is constructed. Varactor diodes are mounted on the top and bottom layers. Due to the bias network design, the dual-transmission windows are independently regulated by tuning varactor diodes. The geometric parameters are fixed as: the periodical length is *p* = 15 mm, the outside lengths of the split square ring on the top and bottom layer are *a* = 12 mm and *b* = 5.4 mm, the widths of thin metal lines are *w*_1_ = 0.3 mm, *w*_2_ = 0.2 mm, and *w*_3_ = 0.3 mm, the thickness of the substrate *h* = 0.6 mm, the branch length on the bottom layer *l* = 1 mm, the space for mounting varactor diodes is *g* = 1 mm, the widths between two adjacent unit cells on the top and bottom layer are *s* = 0.3 mm and *e* = 0.1 mm.

The performance of the designed meta-device is analyzed with the commercial electromagnetic simulation software of CST Microwave Studio. The boundaries were set to unit cell in the x and y directions, open (add space) in the z direction, the frequency-domain solver is chosen to perform the operation. The transmission spectra of the meta-device are plotted in Figure 3.

*C_v_*_1_ and *C_v_*_2_ are capacitances of varactor diodes on the top and bottom layer. When the *C_v_*_1_, *C_v_*_2_ of varactor diodes are 0.35 pF and 0.466 pF, the two resonant frequencies (*f*_1_, *f*_2_) of this meta-device are 2.72 GHz and 4.83 GHz. The corresponding insertion losses are 0.50 dB and 1.54 dB, respectively. The resonance frequency *f*_1_ moves to low frequency from 2.72 GHz to 2.20 GHz when the value of *C_v_*_1_ alters from 0.35 pF to 3.2 pF and the value of *C_v_*_2_ is 0.466 pF. The insertion losses are from 0.50 dB to 0.96 dB. When the value of *C_v_*_2_ changes from 0.466 pF to 2.35 pF and the value of *C_v_*_1_ is 0.35 pF, the resonance frequency *f*_2_ moves to low frequency from 4.83 GHz to 4.27 GHz with the insertion losses from 1.54 dB to 2.28 dB. It can be clearly seen that the resonance frequencies (*f*_1_, *f*_2_) keep shifting when the capacitance values (*C_v_*_1_, *C_v_*_2_) of the varactor diodes increase. The varactor diodes can cause red shifts of the resonant frequencies (*f*_1_, *f*_2_).

As shown in Figure 4, the ECM of the proposed meta-atom is obtained. On the top layer, *C_t_*_1_ and *C_t_*_2_ represent the gap capacitances that are resulted from the couplings of square ring, bias line, and adjacent cells. *L_t_*_1_ and *L_t_*_2_ are the equivalent inductances of the metal strip and split square ring. The metal grid of the middle layer is equated to the inductance *L*_m_. The gap capacitance *C_b_* of adjacent cells and the equivalent inductance *L*_b_ of the metal strip constitute the ECM of the metal pattern on the bottom layer. In addition, *C_v_*_1_ and *C_v_*_2_ are capacitances of varactor diodes on the top and bottom layer. The top metal structure and the metal grid form a parallel L-C resonance (*f*_1_). The metal structures on the bottom layer and the metal grid constitute another parallel L-C resonance (*f*_2_). Based on the proposed ECM in Figure 4, the resonant frequencies (*f*_1_, *f*_2_) are:(3)$$f_{1}=\frac{1}{2\pi \sqrt{\left(C_{t1}+C_{t2}+C_{v1}\right)\left(L_{t1}+L_{t2}+L_{m}\right)}},$$
(4)$$f_{2}=\frac{1}{2\pi \sqrt{\left(C_{b}+C_{v2}\right)\left(L_{b}+L_{m}\right)}}.$$

In order to further demonstrate the principle of the proposed structure, the parametric analysis is discussed and summarized in Figure 5. “a” and “b” are the widths of the split square rings on the top and bottom layers, respectively. They affect the corresponding series L-C resonance in the equivalent circuit. As shown in Figure 5a,d, the corresponding series resonant frequency shift toward higher frequency as they increase. The widths (“s”) between two adjacent unit cells on the top layer affect both the series and parallel resonant frequencies of the corresponding layer in Figure 5b. The metal grid of the middle layer is essential to form the two transmission windows. It can be clearly seen from Figure 5c that as the width of the metal (“w”) increases, the two resonant frequencies move simultaneously to high frequencies. In addition, the thickness of the substrate (“h”) has no significant effect on the resonant frequency in Figure 5e. However, it has an attenuating effect on the transmission coefficient. This is determined by the specific impedance of the substrate used. Therefore, the thick substrate is not suitable for the proposed structure.

When the EM wave illuminate obliquely, the equivalent capacitance and inductance corresponding to metal structures change. This leads to blue shifts of the resonant frequencies (*f*_1_, *f*_2_). By adjusting the equivalent capacitances (*C_v_*_1_, *C_v_*_2_) of varactor diodes, the overall impedance of the ECM is equal to that under normal incidence. The angular dispersion can be compensated in this active method. However, the compensation range of angular dispersion is limited due to the variable capacitance and cut-off voltage of the varactor diodes used. Here, transmission characteristics with incidence angles from 0° to 60° are investigated and analyzed. The resonance frequency *f*_1_ moves to a high frequency (from 2.72 GHz to 3.19 GHz) and the CST simulation results are depicted in Figure 6a. The insertion losses are from 0.50 dB to 0.96 dB. Additionally, the resonance frequency *f*_2_ moves to high frequency from 2.72 GHz to 3.19 GHz with the insertion loss from 1.54 dB to 2.25 dB. Obviously, the effect of angular dispersion becomes intense as the angle of incidence increases. The angular dispersion leads to blue shifts of dual-transmission windows. The blue shifts of angular dispersion and the red shifts introduced by the varactor diodes are in reverse trend. Therefore, the angular dispersion can be compensated by tuning varactor diodes.

Then, the capacitance (*C_v_*_1_, *C_v_*_2_) of varactor diodes is adjusted to compensate the angular dispersion with the incident angle from 0° to 60°. The simulated results for tunable angular dispersion are shown in Figure 6b. When *C_v_*_1_ and *C_v_*_2_ are 0.35 pF and 0.466 pF, the resonance frequencies are 2.72 GHz and 4.83 GHz at vertical incidence. As the incident angle changes to 15°, the resonance frequency *f*_1_ shift to 2.76 GHz. Then, it can remain at 2.72 GHz to tune *C_v_*_1_ at 0.37 pF. The resonance frequency *f*_1_ shift to 2.85 GHz when the incident angle is 30°, which can be converted to 2.72 GHz by changing *C_v_*_1_ at 0.47 pF. At the incidence angle of 45°, the resonance frequency *f*_1_ is 3.01 GHz. In order to remain at 2.72 GHz, the capacitance of *C_v_*_1_ is varied to 0.76 pF. When the incident angle is 60°, the resonance frequency *f*_1_ shift to 3.19 GHz. Then it can be compensated at 2.72 GHz by tuning *C_v_*_1_ at 2.50 pF. In conclusion, the resonance frequency *f*_1_ of the transmission window at 2.72 GHz can be stable by tuning *C*_v1_ when the incidence angle alters from 0° to 60°.

It is also revealed from Figure 6b that the resonance frequency *f*_2_ can remain at 4.83 GHz as the incidence angle changes from 0° to 60°. When the incident angle is 15° and 30°, both the resonance frequency *f*_2_ are 4.83 GHz with *C_v_*_2_ of 0.48 pF and 0.54 pF, respectively. However, the capacitance *C_v_*_2_ of the varactor diode is, respectively, altered to 0.64 pF and 0.9 pF to compensate angular dispersion at 4.83 GHz when the incidence angle is 45° and 60°. The variable capacitances (*C_v1_*, *C_v_*_2_) of varactor diodes to eliminate the angular dispersion with the incident angle from 0° to 60° are shown in Table 1. The above simulation results demonstrate that the blue shifts of angular dispersion can be dynamically compensated by the red shifts introduced by the varactor diodes with the incident angle from 0° to 60°.

## 4. Experimental Verification

A prototype of the proposed meta-device has been fabricated and measured in an anechoic chamber to verify the feasibility of this dynamic modulation method. It has 2916 varactor diodes and 27 × 27 structural cells. Including external bias network, the whole sample size is 525 mm × 525 mm. The substrates used are two 0.6 mm thick F4B. The relative permittivity and loss tangent angles are 2.65 and 0.001, respectively. The surface appearance of the prototype is shown in Figure 7a. The varactor diodes SMV2020-079LF and SMV1231-079LF are embedded in the split square rings on the top and bottom layer, respectively. They are mounted in the same direction. The specific test environment is shown in Figure 7b. Two horn antennas are used to transmit and receive EM waves. The network analyzer used is the Agilent N5224A vector network analyzer. Two DC voltage sources are connected to the top and bottom layers of the prototype to drive the varactor diodes.

The measured results are shown in Figure 8 and Figure 9. It is clearly revealed from Figure 8 that the varactor diodes on the top and bottom layer can regulate the shifts of the dual-transmission windows independently. When *C_v_*_1_ changes from 0.35 pF to 3.20 pF and *C_v_*_2_ is kept at 0.466 pF, the resonance frequency *f*_1_ moves from 2.74 GHz to 2.21 GHz, and the resonance frequency *f*_2_ approximately remains at 4.85 GHz. Conversely, when *C*_v2_ varies and *C_v_*_1_ is constant, the resonance frequency *f*_2_ changes from 4.85 GHz to 4.30 GHz and the resonance frequency *f*_1_ approximately remains at 2.74 GHz.

The excellent performance of the tunable angular dispersion is also demonstrated in Figure 9. In Figure 9a, with the capacitance *C_v_*_1_ of 0.35 pF and *C_v_*_2_ of 0.466 pF, the resonance frequency *f*_1_ shifts from 2.74 GHz to 3.19 GHz and the resonance frequency *f*_2_ shifts from 4.85 GHz to 5.12 GHz when the incidence angle from 0° to 60°. According to the shifting trend of the resonance frequencies, the angular dispersion can be adjusted by tuning *C_v_*_1_ and *C_v_*_2_ simultaneously, which is depicted in Figure 9b. The equivalent capacitance (*C_v_*_1_, *C_v_*_2_) of the varactor diodes and the external DC voltages (*U*_1_, *U*_2_) are shown in Table 2. The capacitance values *C_v_*_1_ on the top layer are tuned to 0.37 pF, 0.47 pF, 0.76 pF and 2.50 pF at incident angle of 15°, 30°, 45°, and 60°, in which the external DC voltages *U*_1_ are at 15.1 V,12.1 V, 7.5 V, and 0.8 V. The resonance frequency *f*_1_ approximately remains at 2.73 GHz from 0° to 60°. At the same time, the capacitance *C*_v2_ also need tune at 15°, 30°, 45°, and 60°, which are 0.48 pF, 0.54 pF, 0.64 pF, and 0.90 pF with relative to 9.4 V, 7.9 V, 6.2 V, and 3.5 V of the DC voltages *U*_2_. All of these experimental results agree reasonably well with the simulations. However, due to the measured environment and manufacturing uncertainty, there are still some deviations in the measuring results. The parameters of the varactor diodes are not constant such as in the datasheet in the actual measurement, which also interferes with the test results. In addition, there are some undulations in measurement results. The output of the signal transmitter through the power amplifier is not stable. The free-space method itself has this limitation. The wave generated from the transmitting horn is also not strictly planar. This can also produce fluctuations in the test results.

The performance comparison of the meta-devices with dual transmission windows is shown in Table 3. Passive meta-devices can only weaken the effect of angular dispersion from their structure. They are still limited by the angular range and angular dispersion. The active meta-device can shift the resonant frequency of the transmission window by introducing active components. However, the angular dispersion still has an impact on the EM performance as the incident angle increases. The proposed method for compensating angular dispersion generates a red shift by adjusting the varactor diodes. At oblique incidence, the red shift is compensated with the blue shift of the angular dispersion. Thus, the proposed meta-device achieves the characteristic of tunable angular dispersion. This advancement makes it have great application prospect in angle-multiplexed communications, and incident-angle-insensitive equipment.

## 5. Conclusions

In this paper, a paradigm for tunable angular dispersion with the varactor diodes is proposed. The main conclusions of this article are as follows:(1)A method to dynamically modulate the angular dispersion is proposed and validated. The varactor diode can continuously modulate the resonant frequency of the transmission window. Thus, the angular dispersion can be dynamically compensated by mounting the varactor diode in the passive meta-atom.(2)The modulation method is applied to meta-devices with dual-transmission windows. Based on the analysis of the equivalent circuit model, the overall impedance of the structure is maintained constant by modulating the varactor diode. Thus, the blue shift of the angular dispersion and the red shift introduced by the varactor diode counteract each other to achieve angular dispersion compensation.(3)As a proof-of-principle, a triple-layer meta-device with dual-transmission windows is designed and analyzed. The dual-transmission windows are, respectively, adjusted by varactor diodes on the top and bottom layer. With CST simulation and measurement, it can achieve the angular dispersion compensation at 2.72 GHz and 4.83 GHz as the incidence angle changes from 0° to 60°.

Therefore, the proposed meta-device provides a method for tunable angular dispersion based on active components, which has applications in angle-multiplexed communications, and incident-angle-insensitive equipment. In addition, it still has great prospects for applications in terahertz and other optical fields.

## Figures and Tables

**Figure 1 materials-15-03686-f001:**
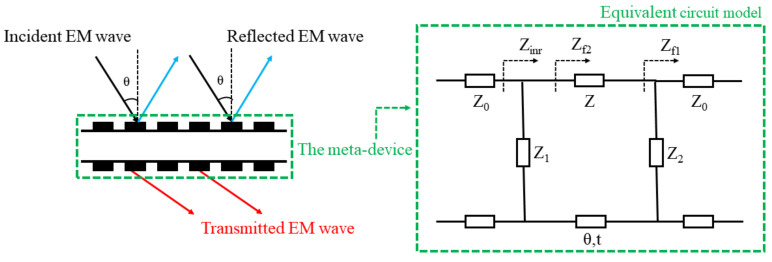
The diagram of the path of action of EM waves and the corresponding equivalent circuit model.

**Figure 2 materials-15-03686-f002:**
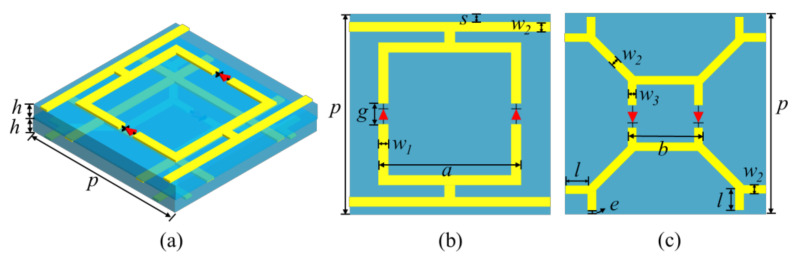
Geometry of the proposed meta-atom. (**a**) 3-D view of the structure. (**b**) The top layer. (**c**) The bottom layer.

**Figure 3 materials-15-03686-f003:**
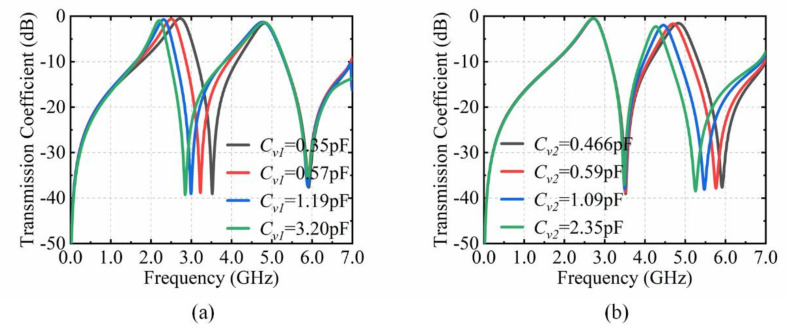
CST simulation results of the proposed structure. (**a**) Transmission coefficient with *C_v_*_1_ from 0.35 pF to 3.20 pF. (**b**) Transmission coefficients with *C_v_*_2_ from 0.466 pF to 2.35 pF.

**Figure 4 materials-15-03686-f004:**
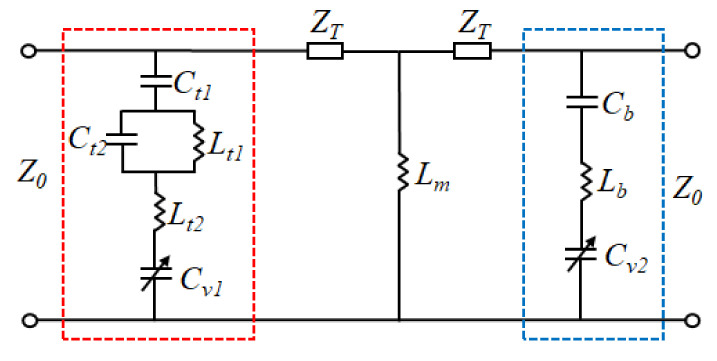
Simplified ECM of the meta-device.

**Figure 5 materials-15-03686-f005:**
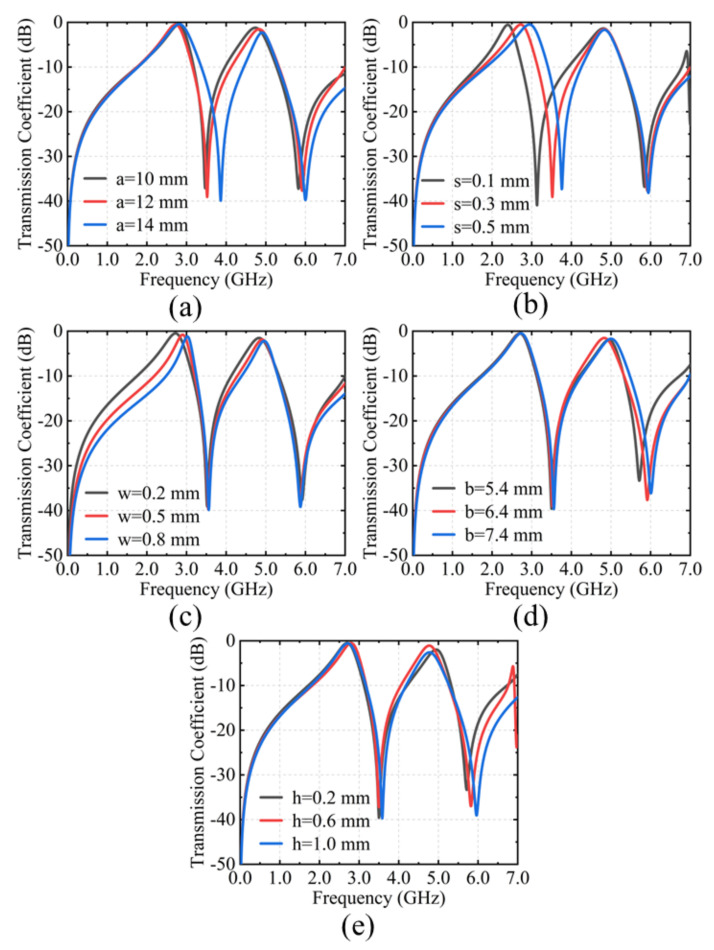
Parametric analysis by CST simulation. (**a**) The outside lengths of the split square ring on the top layer. (**b**) The width between two adjacent unit cells on the top layer. (**c**) The width of the metal grid on the middle layer. (**d**) The outside lengths of the split square ring on the bottom layer. (**e**) The thickness of the substrate.

**Figure 6 materials-15-03686-f006:**
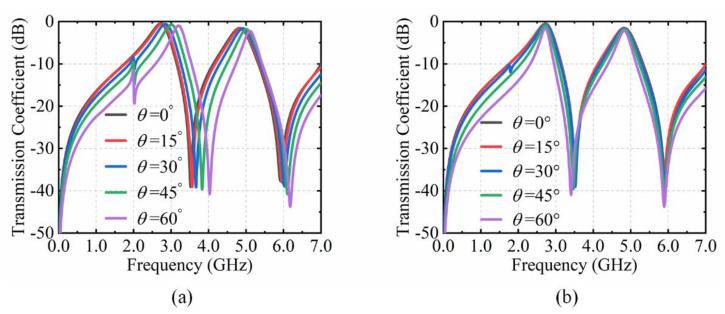
CST simulation results of the proposed structure. (**a**) Transmission coefficient with incidence angles from 0° to 60°. (**b**) The effect of angular dispersion compensation with incidence angles from 0° to 60°.

**Figure 7 materials-15-03686-f007:**
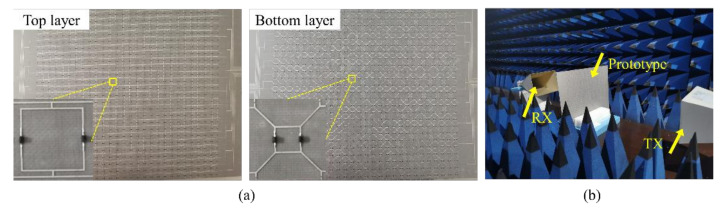
(**a**) The surface appearance of the prototype. (**b**) The specific test environment.

**Figure 8 materials-15-03686-f008:**
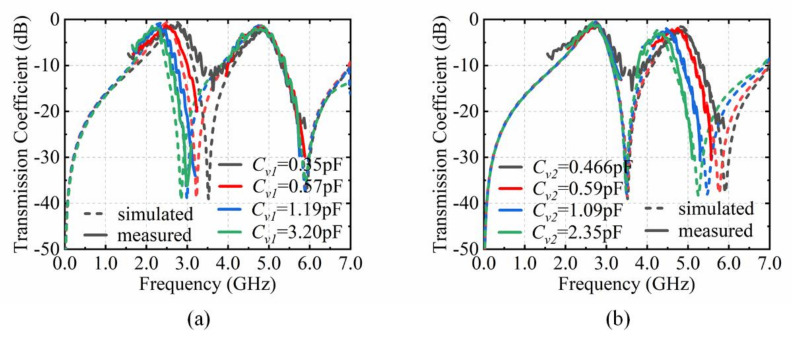
Comparison of measurement results and simulation. (**a**) Transmission coefficient with C*_v_*_1_ from 0.35 pF to 3.20 pF. (**b**) Transmission coefficient with C*_v_*_2_ from 0.466 pF to 2.35 pF.

**Figure 9 materials-15-03686-f009:**
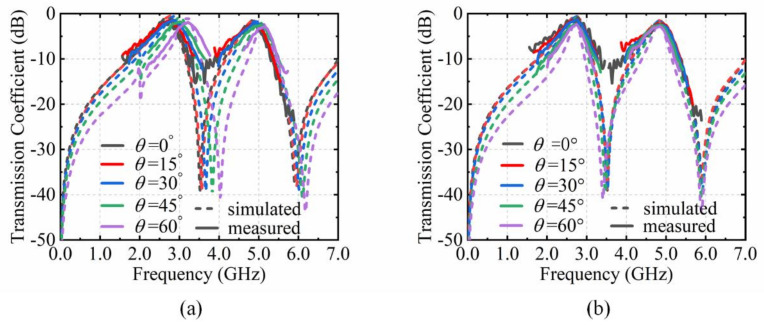
Comparison of measurement results and simulation. (**a**) Transmission coefficient with the incident angle from 0° to 60°. (**b**) Transmission coefficient for angular dispersion compensation by tuning *C_v_*_1_ and *C_v_*_2_.

**Table 1 materials-15-03686-t001:** The variable capacitance of varactor diodes with the incident angle from 0° to 60°.

The Incident Angle (°)	The Variable Capacitance of the Varactor Diode (pF)	
	The Top Layer C_v1_	The Bottom Layer C_v2_
0	0.35	0.466
15	0.37	0.48
30	0.47	0.54
45	0.76	0.64
60	2.50	0.90

**Table 2 materials-15-03686-t002:** The measurement parameters for achieving angular non-dispersion.

The Incident Angle (°)	The Capacitance C_v1_ (pF)	DC Voltage U_1_ (V)	The Capacitance C_v2_ (pF)	DC Voltage U_2_ (V)
0	0.35	16.2	0.466	10.2
15	0.37	15.1	0.48	9.4
30	0.47	12.1	0.54	7.9
45	0.76	7.5	0.64	6.2
60	2.50	0.8	0.90	3.5

**Table 3 materials-15-03686-t003:** Comparison with the meta-devices for dual-transmission windows.

Reference	Type	Dual-Transmission Windows (GHz)	Angular Stability	Angular Dispersion
[30]	passive	27.7/39.1	45°	2.3%
[31]	passive	2.5/5.5	45°	Not stated
[32]	active	2.28–4.66/5.44–11.3	60°	Not stated
[33]	active	0.28–1.28/0.52–1.98	60°	Not stated
Proposed paper	active	2.74/4.85	60°	tunable

## Data Availability

Not applicable.

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
