# Peer review of "Active Meta-Device for Dual-Transmission Windows with Tunable Angular Dispersion Characteristics"

_materials, 2022, doi:10.3390/ma15103686_

Round 1

Reviewer 1 Report

This work represents metasurfaces for metadevices application. 3 layers metasurface is introduced to tune the blue shift and red shift. The work is interesting for scientific community. 

I would ask the authors to add the percentage difference  in simulated and measured results. Also, would be better to shed light on the fine tuning of red and blue shifts with respect to number of layers of metasurface. Any why specifically 0 to 60 degrees range is suggested. Also the metasurface parameters effect with specific frequencies, explain if possible.

Reviewer 2 Report

This work implements an active device with dual transmission window with different angular characteristics in the 2 and 4 GHz bands using varactor diodes and a multilayer structure. The work done is interesting , but needs a major revision before being published in the journal. 

Minor comments:

Include some more reference in the introduction to metasurfaces with varactor diodes. 

The quality of all figures is poor.

include all the parameters of the structure in Figure 2, so that readers can replicate the design. 

 Specific comments:

Why have the 2.72 and 4.83 GHz frequencies been selected in the metasurface design? are they random?  have they been selected for a particular application? 

What would the design of the meta-atom look like at different frequencies?

In figure 2a there is a resonance around 2 GHz, what is the reason for this resonance?

Which network analyser was used to specify?

Include a comparative table with the advances included in this work in comparison with other simulated works, highlighting the advantages of using this type of device? 

The measurements in figures 7 and 8 only appear in the 1.5 to 6 GHz range, could you include measurements over the entire simulation range?

What is the tolerance in the fabrication process of the metasurface? Are the electrical characteristics of the substrate defined by the manufacturer or have they been analysed?

In the measurements in figures 7 and 8 there are some undulations, what could these be due to?

I suggest that in order to be able to better see the simulation and measurement of each angle of incidence they should be represented in different figures?

Reviewer 3 Report

Title: Active meta-device for dual-transmission windows with tunable angular dispersion characteristics

The article proposed a novel method for the attainment of dual-transmis

sion windows concurrently with the elimination of angular dispersions.

The researchers made a detailed assessment based on diagnostic investigation of the dynamic compensation strategy by the varactor diodes with bias voltage. Some statements are made on the adjustment of dual transmission window.  Based on these observations some recommendations are provided. This is good article and must be published if the following minor issues are appropriately adhered to.

  1. In the text of the manuscript the first person like “we” should not be used.
  2. The literature review is not elaborate and a separate dedicated paragraph is needed to state the research gaps. Subsequently the motivation and the objectives of the present research should spontaneously evolve from the research gap. This feature is strikingly missing in the manuscript.
  3. In the literature review the citations should be referred one at a time (not to be cited in clubbed way).
  4. FESEM provides better results and better visualization. Few FESEM images are to be added or the explanation of not incorporating the FESEM images have to be stated categorically.
  5. The discussions in the conclusion portions have to be brief and to the point. All the elaborate discussion should be in the result analysis part not in the conclusion portion.
  6. The statements in the conclusion portion should be bulleted for better comprehensibility.

Reviewer 4 Report

Major Comments:

1) Writing aspects: English language requires a bit of editing since some sentences need to be shorten.

2) Originality: The novelty of the submission and relevance of the contributions are under major question. After a literature survey on the submitted manuscript context, it has been found that the submitted manuscript is very similar to recently published papers by other researchers on Active meta-device for dual-transmission windows. The published papers are even stronger and implemented advanced constitutive models, FEM and AI to replicate their experiments. Introduction must be re-rewritten a bit to show the state of the art and novelty. The following papers can be added to enhance the quality on the shape morphing aspects:

Shape-adaptive metastructures with variable bandgap regions by 4D printing. Polymers, 12(3), p.519.

Resonant THz transmission through asymmetric aperture array with polarization controlled resonant peaks and Q-factors. Journal of Applied Physics, 126(22), p.223103.

3) Clarity of content: How to make sure that dual transmission exists for several cycles? It needs to be studied and shown in the results experimentally and numerically.

4) Band gaps must be shown.

Conclusion:
In view of the above, the manuscript is not recommended for publication in its present form. The authors should modify the manuscript and address the above comments. In my opinion, a revised version including above mentioned points may be considered for publication after reconsideration.

Round 2

Reviewer 2 Report

The authors have responded to my suggestions and comments and the article is ready for publication in the journal, I suggest correcting some spelling mistakes in the text.

Reviewer 4 Report

The comments have been addressed. The manuscript is recommended for publication in its revised form.